# Identifying putative substrates of Calpain-15 in neurodevelopment

**Congyao Zha**[1], **Ally Huang**[2], **Senthilkumar Kailasam**[3], **Daniel Young**[4], **Antoine Dufour**[4], **Wayne S. Sossin**[1]*

1 Department of Neurology and Neurosurgery, Montreal Neurological Institute, McGill University, Montreal, Quebec, Canada, 2 Department of Anatomy and Cell Biology, McGill University, Montreal, Quebec, Canada, 3 Canadian Centre for Computational Genomics, McGill University, Montreal, Quebec, Canada, 4 Department of Physiology and Pharmacology, University of Calgary, Calgary, Canada

* wayne.sossin@mcgill.ca

## Abstract

Calpain 15 (CAPN15) is an intracellular cysteine protease belonging to the non-classical small optic lobe (SOL) family of calpains, which has an important role in developmental processes. Loss of *Capn15* in mice leads to developmental eye anomalies and volumetric changes in the brain. Human individuals with biallelic variants in *CAPN15* have developmental delay, neurodevelopmental disorders, as well as congenital malformations, including eye anomalies. However, the substrates of Capn15 are still unidentified. Here, using *Capn15* KO P2 mice of both sexes, we have used RNA sequencing (RNA-SEQ), proteomics, and N-terminomics/terminal amino isotopic labelling of substrates (TAILS), to examine putative substrates of Capn15. There were few changes in the transcriptome profile, and we could not verify a protein change in one selected mRNA between *Capn15*-/- and WT mice, although a putative transcription factor linked to these changes, Pax2, did show a significant increase after the loss of Capn15. TAILS revealed a preference for cleavage at basic residues, and while no hits showed a significant change in cleavage, some were more abundant when Capn15 was removed. These included Doublecortin and Tubb3, and the Doublecortin predicted cleavage was at a lysine residue. Cleavages at lysine residues were enriched in peptides that were lost or reduced when Capn15 was removed, but not in cleavages that were unchanged when Capn15 was removed.

## Introduction

Calpains are intracellular cysteine proteases and consist of four conserved families:. Classical, PalB Transformer and Small Optic Lobe (SOL). The SOL family is defined by N-terminal zinc fingers and a C-terminal SOL domain [1]. Here we focus on the sole SOL calpain in mammals, Capn15. The N-terminal zinc fingers of SOL calpain binds polyubiquitin [2] but binding partner of C-terminal SOL homology domain have not been identified [1,3]. This family was first identified in the fruit fly *Drosophila* [4,5] where loss of the SOL calpain leads to smaller optic lobes [5]. Recently, we found that removal of Capn15 in rodents leads to a smaller mendelian ratio, lighter weights, and neurodevelopmental issues including smaller brains and eye anomalies [6,7]. Humans carrying homozygous variants of *CAPN15* also exhibited a variety of neurodevelopmental disorders [7–10].

**Data availability statement:** All immunoblots are added as a supplemental file All RNA seq data is available from the GEO repository GSE287980 All proteomic data is available from the PRIDE repository PXD060034.

**Funding:** CIHR grant 340328 to WSS. CIHR stands for Canadian Institutes for Health Research. The funders had no role in study design, data collection and analysis, decision to publish, or preparation of the manuscript.

**Competing interests:** The authors have declared that no competing interests exist.

In the adult mouse, loss of Capn15 in forebrain excitatory neurons leads to some behavioral deficits including decreased fear generalization [11]. In *Aplysia*, dominant negative forms of SOL calpain block formation of non-associative forms of plasticity [12,13]. While these results suggest some role for Capn15 in adult plasticity, Capn15 is expressed at much higher levels during early neuronal development than in the adult [7] and here, we solely examine targets of Capn15 during neurodevelopment.

Calpains can regulate transcription through transcription factors both directly and indirectly during development [14–17]. Thus, we examined whether there were changes in transcription using RNA-SEQ. We found that the levels of two transcription factors linked to changes seen during RNA-SEQ, Pax2 and Pax 5 are significantly increased in Capn15 KO. N-terminomics/TAILS is a technique to examine differences in internal N-terminal sequences and has been used to identify substrates (targets) of numerous proteases [18,19]. The TAILS protocol also allows evaluation of proteomic changes regardless of cleavage products. We have used all of these to identify Capn15 targets comparing KO animals to the phenotypically normal heterozygotes. Putative targets were identified and we attempted verification by immunoblotting. We identified a number of proteins whose levels were increased after loss of Capn15, but we found no evidence for the loss of any cleavage product by immunoblotting for any of the putative targets examined.

## Materials and methods

### Animals

All animal experiments were carried out in compliance with protocols approved by the Montreal Neurologic Institute animal committee (Protocol MNI-5784). Adult mice were anesthetized with Isoflurane and $CO_2$ followed by sacrifice with cervical dislocation. Neonates were sacrificed by decapitation. C57BL/6J mice were used to generate our mouse lines. The generation of the *Capn15*$^{(lacZ-Neo)}$ mouse and the FLOXed Capn15 mouse has been described elsewhere [7].

### RNA sequencing

Brains were isolated from P2 animals and stored at −80 °C.14 Frozen brains from 6 litters (7 WT and 7KO) were used for RNASEQ. RNA library construction, sequencing and bioinformatics analysis were performed by the McGill University and Genome Quebec Innovation Centre. rRNA depletion was performed on 200–400 ng of each total RNA sample with the RiboZeroGold (Illumina, San Diego, CA, USA) as per the manufacturer's instructions. The entire rRNA depleted fraction (ranging 4–22 ng) was used as input for library preparation using the ScriptSeq V2 library preparation kit (Illumina, San Diego, CA, USA). All libraries were validated and quantified with the Bioanalyzer DNA 1000 assay (Agilent Technologies, Inc., CA, USA) and further quantified with the Qubit DNA Broad Range assay (Life Technologies, Carlsbad, CA, USA). 10 μL of each library were diluted to a concentration of 10 nM. Equal volumes of each 10 nM library were then pooled for subsequent paired-end sequencing on an Illumina HiSeq 2000/2500 (Illumina, San Diego, CA, USA). Sequencing was performed with 4 samples per lane, hence generating 62–106 million paired end reads per library. Base calls were made using the Illumina CASAVA pipeline. Base quality was encoded in phred 33.

Sequence alignment and quantification of gene expression were carried out using an internally developed RNA-Seq analytical pipeline [20].Reads were trimmed from the 3′ end to have a phred score of at least 30. Trimming was done with the Trimmomatic software [21]. The filtered reads were aligned to a reference genome GRCm38. The alignment was done with STAR 2.7.8 software [22]. Reads association with annotated gene regions was done using the

HTseq-count tool v0.11.1 [23]. Differential expression analysis was performed with edgeR Bioconductor package [24]. Gene ontology (GO) and Kyoto Encyclopedia of Genes and Genomes (KEGG) enrichment analyses was implemented using the clusterProfiler package [25]. All RNA-SEQ data is available at the GEO database. Geo submission ID is GSE287980

## N-terminomics/terminal isotopic labeling of substrates (TAILS) and shotgun proteomics

Brains were harvested from P2 *Capn15*[+/-] (CAPN15 het) and *Capn15*[-/-] (CAPN15 KO) mice (n = 5) and homogenized. Brain homogenate from both *Capn15* het and KO mice were treated with 6M guanidine HCl (pH 8.0) and subjected to an N-terminomics/TAILS and shotgun proteomics workflow [26,27]. Samples were reduced with 5 mM DTT (Gold Biotechnology, St-Louis, MO) at 37 °C for 1 h and alkylated with 15 mM IAA (GE Healthcare, Mississauga, ON) in the dark at room temperature for 30 min followed by quenching with 15 mM DTT. The pH was adjusted to 6.5 before the samples were isotopically labelled with either a final concentration of 40 mM deuterated heavy formaldehyde ($^{13}CD_2O$) or 40mM light formaldehyde ($^{12}CH_2O$) in presence of 40 mM sodium cyanoborohydride overnight at 37 °C. Next, samples were combined and were precipitated using acetone/methanol (8:1). The resulting pellet was resuspended in 1M NaOH and the proteins were subjected to trypsin (Promega, Madison, WI) digestion overnight at 37 °C. For pre-enrichment TAILS (pre-TAILS)/shotgun proteomics, 10% of the trypsin-digested samples were collected and the pH was adjusted to 3 with 100% formic acid. The rest of the samples were adjusted to a pH of 6.5 and incubated with a 3-fold excess (w/w) of dendritic polyglycerol aldehyde polymer overnight at 37 °C. Unbound peptides from the polymer-bound peptides were filtered out by centrifugal filter unit with 10-kDa cut-off membrane (Amicon Ultra, Millipore) at 10,000 *g* for 5 min. The flow through was collected and the Amicon columns were washed with 100mMTris-HCl, pH 6.5. The pH of the samples was adjusted to 3 with 100% Formic acid. Both pre-TAILS and TAILS samples were then desalted using Sep-Pak C18 columns and lyophilized before submitting for LC-MS/MS analysis to the Southern Alberta Mass Spectrometry core facility, University of Calgary, Canada.

## High-performance liquid chromatography (HPLC) and mass spectrometry

All liquid chromatography and mass spectrometry experiment were carried out by the Southern Alberta Mass Spectrometry core facility at the University of Calgary, Canada. Analysis was performed on an Orbitrap Fusion Lumos Tribrid mass spectrometer (Thermo Fisher Scientific, Mississauga, ON) operated with Xcalibur (version 4.0.21.10) and coupled to a Thermo Scientific Easy-nLC (nanoflow Liquid Chromatography) 1,200 system. Tryptic peptides (2 µg) were loaded onto a C18 trap (75 µm× 2 cm; Acclaim PepMap 100, P/N 164946; ThermoFisher Scientific) at a flow rate of 2 µL/min of solvent A (0.1% formic acid and 3% acetonitrile in LC-mass spectrometry grade water). Peptides were eluted using a 120 min gradient from 5 to 40% (5% to 28% in 105min followed by an increase to 40% B in 15min) of solvent B (0.1% formic acid in 80% LC-mass spectrometry grade acetonitrile) at a flow rate of 0.3% µL/min and separated on a C18 analytical column (75 µm× 50 cm; PepMap RSLC C18; P/N ES803; Thermo Fisher Scientific). Peptides were then eletrosprayed using 2.3 kV into the ion transfer tube (300 °C) of the Orbitrap Lumos operating in positive mode. The Orbitrap first performed a fullmass spectrometry scan at a resolution of 120, 000 FWHM to detect the precursor ion having a mass-to-charge ratio (m/z) between 375 and 1,575 and a +2 to +4 charge. The Orbitrap AGC (Auto Gain Control) and the maximum injection time were set at $4 \times 10^5$ and 50 ms, respectively. The Orbitrap was operated using the top speed mode with a 3 s cycle

time for precursor selection. The most intense precursor ions presenting a peptidic isotopic profile and having an intensity threshold of at least $2 \times 104$ were isolated using the quadrupole (isolation window of m/z 0.7) and fragmented with HCD (38% collision energy) in the ion routing Multipole. The fragment ions (MS2) were analyzed in the Orbitrap at a resolution of 15,000. The AGC, the maximum injection time and the first mass were set at $1 \times 10^5$, 105 ms, and 100 ms, respectively. Dynamic exclusion was enabled for 45 s to avoid of the acquisition of the same precursor ion having a similar m/z ($\pm 10$ ppm).

## Proteomic data and bioinformatic analysis

Spectral data were matched to peptide sequences in the mouse UniProt protein database using the MaxQuant software package v.1.6.0.1, peptide-spectrum match false discovery rate (FDR) of <0.01 for the shotgun proteomics data and <0.05 for the N-terminomics/TAILS data. Search parameters included a mass tolerance of 20 p.p.m. for the parent ion, 0.05 Da for the fragment ion, carbamidomethylation of cysteine residues (+57.021464), variable N-terminal modification by acetylation (+42.010565 Da), and variable methionine oxidation (+15.994915 Da). For the shotgun proteomics data, cleavage site specificity was set to Trypsin/P (search for free N-terminus and only for lysines), with up to two missed cleavages allowed. For the N-terminomics/TAILS data, the cleavage site specificity was set to semi-ArgC (search for free N-terminus) for the TAILS data and was set to ArgC for the preTAILS data, with up to two missed cleavages allowed. Significant outlier cut-off values were determined after log(2) transformation by boxplot-and-whiskers analysis using the BoxPlotR tool. Database searches were limited to a maximal length of 40 residues per peptide. Peptide sequences matching reverse or contaminant entries were removed.

Uniprot, TopFINDer [28], and Icelogo [29] are used to interpret and analyse data from proteomics. All proteomic data is available at the PRoteomics IDEntifications database (PRIDE), **Project accession:** PXD060034.

## Power analysis

We used the standard deviation of control heterozygote immunoblots. Based on this standard deviation of 0.66 and a power analysis using the variance of $\alpha = 0.05$ and power of 80%, an n of 14 was selected. This n was used for all experiments, although occasionally for technical reasons, the n was 13 (see Figure legends).

## Immunoblotting

Brains were harvested from P2 mice and homogenized manually in lysis buffer containing 25 mM Tris-HCl (pH 7.4), 150 mM NaCl, 6 mM $MgCl_2$, 2 mM EDTA, 1.25% NP-40, 0.125% SDS, 25 mM NaF, 2 mM $Na_4P_2O_7$, 1 mM dithiothreitol (DTT), 1 mM phenylmethylsulfonyl fluoride (PMSF), 20 mg/ml leupeptin, and 4 mg/ml aprotinin. Before loading, 5X sample buffer was added to the lysate and samples were incubated at 95°C for 5 min. Proteins were resolved by SDS-PAGE on Bis-Tris gel and transferred to nitrocellulose membrane (Bio-Rad). The blots were blocked in TBST (TBS + 0.1% Tween) containing 5% skim milk for 30 min at room temperature and then incubated with primary antibodies at either room temperature for 1 hr or 4°C overnight. After washing 3 times with TBST, the blots were incubated with HRP-conjugated secondary antibodies for 1 h at room temperature and washed again 3 times in TBST. The Western Lightning Plus- ECL kit (NEL103001EA; PerkinElmer LLC Waltham, MA USA) was used as per manufacturer's instructions to detect protein bands.

In most cases, we chose antibodies for which there were published evidence for loss of immunoreactivity in immunoblotting after knock out. For the primary antibody,

we used rabbit polyclonal antibody directed against autotaxin Enpp2 (Proteintech #14243–1-AP, 1:1000); rabbit polyclonal antibody directed against Smarca4/Brg1 (Abcam #110641, 1:1000) [30]; rabbit polyclonal antibody directed against Pax2 (Abcam #79389, 1:1000); rabbit-monoclonal antibody against Pax 5 (Cell signalling technology #12709, 1:250 [31]mouse monoclonal antibody against Runx2 (gift from Stefano Stifani); rabbit polyclonal antibody directed against Dhx9 (Proteintech #17721–1-AP, 1:1000) [32]; rabbit polyclonal antibody directed against Rbfox2 (Sigma-Aldrich HPA006240, 1:500) [33]; rabbit polyclonal antibody directed against Ctnnb1 (Proteintech #51067–2-AP, 1:1000) [34]; rabbit polyclonal antibody directed against Tubb3 (ABclonal #A17074, 1:1000); rabbit polyclonal antibody directed against eEF2 (Cell signalling technology #2332S, 1:1000); rabbit polyclonal antibody directed against Crmp1 (Abcam #199722, 1:1000) [35]; rabbit polyclonal antibody directed against histone H2A (Cell signalling technology #2578, 1:250); rabbit polyclonal antibody directed against histone H4 (Abcam #10158, 1:750); mouse monoclonal antibody directly against Dcx (Santa Cruz sc-271390, 1:500) [36]. The secondary antibody was horseradish peroxidase-conjugated goat anti-rabbit/mouse secondary antibody (1:5000). Antibodies were diluted in Tris buffered saline with Tween containing 5% skim milk powder. All immunoblots can be viewed in S1 Supplemental Data.

## Quantification of immunoblots

Immunoblots were scanned and imaged using the public domain Image J program developed at the U.S. National Institute of Health (https://imagej.nih.gov/ij/). We calibrated our data with the uncalibrated optical density feature of NIH image, which transforms the data using the formula $\log_{10}\left[\dfrac{225}{225-x}\right]$, where x is the pixel value (0–254). We used the Ponceau image for each gel to normalize the amount of protein of interest. For each gel, we divided all values by the average of the normalized values to be able to compare data across multiple gels. Data were analysed with Students t-test.

## Results

For all experiments, we used P2 brains, as Capn15 is widely distributed in the brain during this period of development [7]. We first investigated genes that are differentially expressed between the Capn15 KO and the control using RNA-SEQ (S2 supplemental data). When sex and litter were included as variables, there were 38 mRNAs with significant changes in abundance with a differential gene expression (DGE) analysis and FDR cut off of 0.05 (Fig 1A; Table 1). 30 of the 38 mRNAs showed downregulation in the absence of Capn15 (Fig 1A; Table 1). Go analysis showed a preference for mRNAs with the collagen-containing extracellular matrix (Table 2). We examined the protein levels of one putative extracellular matrix target but did not see a significant change in the protein levels of Enpp2 in P2 brains (Fig 1B). Since our hypothesis was that changes in RNA-SEQ would be downstream of Capn15 regulation of a transcription factor, we analyzed putative transcription factors that could regulate the list of differential mRNAs with LISA analysis [37]. We tested four of the hits from this screen (S1 ExtendedData Table, S2 Extended Data Table): Pax2, which is known to be a target for mutations that cause similar ocular and kidney phenotypes as the loss of Capn15 [38,39], Pax5, where haploinsufficiencies cause neurodevelopmental disorders [40], Runx2, a transcription factor implicated in bone development also may play a role in developmental patterning [41]and Smarca4, a component of the mammalian SWI/SNF complex linked to neurodevelopmental disorders such as Coffin Sirius syndrome [42] and eye development

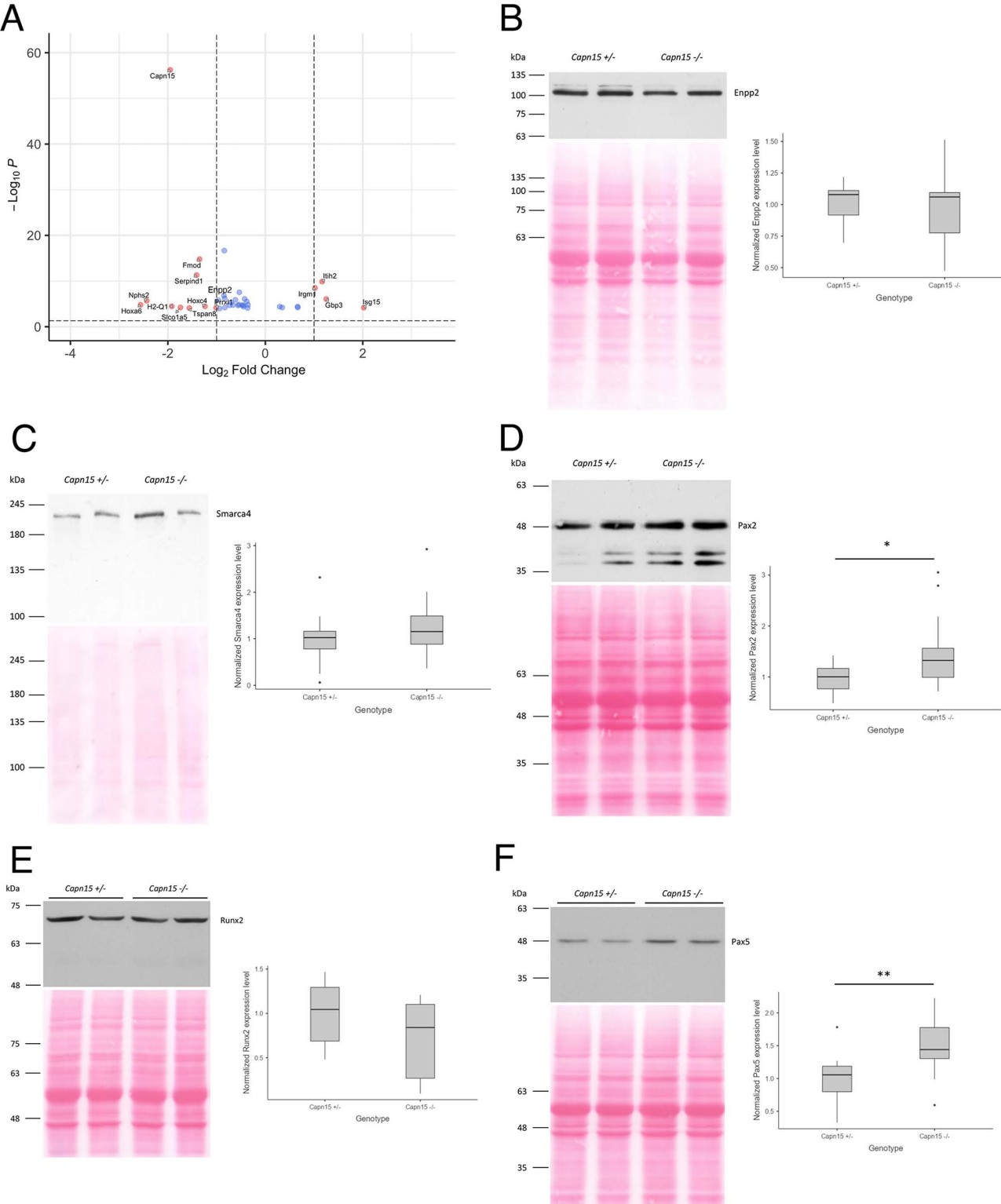

**Fig 1. Verification of RNASEQ targets with western blots.** (A) Volcano plot showing fold changes (x axis) and false discover rate (Y axis). Red circles Top, genes that are differentially expressed between the two genotypes. Only genes with an FDR < 0.05 are shown (blue circles). Genes that in addition have a fold change >2 are circled in red Bottom, Gene Ontology (GO) analysis of genes that are downregulated in the *Capn15-/-* mice. n = 7 for wild type mice; n = 7 for *Capn15-/-* mice. (B) Left, Western blot of P2 mouse brain homogenate targeting Enpp2. Right, Box and whisker plot

showing the normalized quantification of Enpp2 in *Capn15*-/- mice (t-test, *p* = 0.89). (C) Left, Western blot of P2 mouse brain homogenate targeting Smarca4. Right, Box and whisker plot showing the normalized quantification of Smarca4 in *Capn15*-/- mice (t-test, *p* = 0.21). (D) Left, Western blot of P2 mouse brain homogenate targeting Pax2. Right, Box and whisker plot showing the normalized quantification of Pax2 in *Capn15*-/- mice (t-test, *p* = 0.03). n = 13 for *Capn15*+/- mice; n = 14 for *Capn15*-/- mice. Dots are data points outside the box and whiskers. \**p* < 0.05.

**Table 1. Genes that are differentially expressed between the two genotypes. Only genes with an FDR < 0.05 are shown.. n = 7 for wild type mice; n = 7 for *Capn15*-/- mice.**

| Gene | Symbol | logFC | PValue | FDR |
|---|---|---|---|---|
| mRNA that are decreased after loss of Capn15 | | | | |
| ENSMUSG00000037326 | Capn15 | −1.953606232 | 5.75E-57 | 1.03E-52 |
| ENSMUSG00000015090 | Ptgds | −0.84268692 | 2.16E-17 | 1.94E-13 |
| ENSMUSG00000041559 | Fmod | −1.350556038 | 1.64E-15 | 9.82E-12 |
| ENSMUSG00000022766 | Serpind1 | −1.410852074 | 5.01E-12 | 2.25E-08 |
| ENSMUSG00000061808 | Ttr | −0.533154023 | 2.94E-08 | 7.54E-05 |
| ENSMUSG00000021390 | Ogn | −0.850261686 | 1.50E-07 | 0.000335327 |
| ENSMUSG00000061878 | Sphk1 | −0.834302332 | 5.83E-07 | 0.001161821 |
| ENSMUSG00000020396 | Nefh | −0.426022557 | 8.20E-07 | 0.001430341 |
| ENSMUSG00000039728 | Slc6a5 | −0.614104298 | 1.29E-06 | 0.001933069 |
| ENSMUSG00000051029 | Serpinb1b | −0.964822695 | 1.57E-06 | 0.00216715 |
| ENSMUSG00000026602 | Nphs2 | −2.434508879 | 2.03E-06 | 0.00259384 |
| ENSMUSG00000022594 | Lynx1 | −0.381198864 | 2.96E-06 | 0.003535547 |
| ENSMUSG00000018459 | Slc13a3 | −0.702204636 | 4.60E-06 | 0.005148769 |
| ENSMUSG00000047793 | Sned1 | −0.544800447 | 1.37E-05 | 0.014489881 |
| ENSMUSG00000043219 | Hoxa6 | −2.560691072 | 1.58E-05 | 0.015212403 |
| ENSMUSG00000022425 | Enpp2 | −0.356617181 | 1.61E-05 | 0.015212403 |
| ENSMUSG00000031465 | Angpt2 | −0.661954272 | 1.81E-05 | 0.015409639 |
| ENSMUSG00000056174 | Col8a2 | −0.477024803 | 1.80E-05 | 0.015409639 |
| ENSMUSG00000048763 | Hoxb3 | −0.732277094 | 2.04E-05 | 0.016166659 |
| ENSMUSG00000057722 | Lepr | −0.558463687 | 2.16E-05 | 0.016166659 |
| ENSMUSG00000055653 | Gpc3 | −0.446850614 | 2.13E-05 | 0.016166659 |
| ENSMUSG00000079507 | H2-Q1 | −1.920342171 | 3.33E-05 | 0.02363722 |
| ENSMUSG00000075394 | Hoxc4 | −1.238266487 | 3.43E-05 | 0.02363722 |
| ENSMUSG00000026452 | Syt2 | −0.449676763 | 3.97E-05 | 0.024542064 |
| ENSMUSG00000063975 | Slco1a5 | −1.742101675 | 5.87E-05 | 0.033323136 |
| ENSMUSG00000020473 | Aebp1 | −0.828624655 | 6.01E-05 | 0.033323136 |
| ENSMUSG00000041730 | Prrxl1 | −1.011258456 | 6.87E-05 | 0.035204165 |
| ENSMUSG00000036103 | Colec12 | −0.364520291 | 7.70E-05 | 0.038329617 |
| ENSMUSG00000034127 | Tspan8 | −1.559870458 | 8.04E-05 | 0.038938368 |
| ENSMUSG00000030109 | Slc6a12 | −0.938801723 | 0.000102585 | 0.048395798 |
| mRNA that are increased after loss of Capn15 | | | | |
| ENSMUSG00000037254 | Itih2 | 1.165397453 | 1.33E-10 | 4.76E-07 |
| ENSMUSG00000046879 | Irgm1 | 1.020479243 | 3.16E-09 | 9.43E-06 |
| ENSMUSG00000028268 | Gbp3 | 1.249754677 | 8.78E-07 | 0.001430341 |
| ENSMUSG00000038248 | Sobp | 0.294853526 | 3.97E-05 | 0.024542064 |
| ENSMUSG00000054404 | Slfn5 | 0.663319615 | 3.80E-05 | 0.024542064 |
| ENSMUSG00000025789 | St8sia2 | 0.346045685 | 6.05E-05 | 0.033323136 |
| ENSMUSG00000070327 | Rnf213 | 0.664903424 | 6.13E-05 | 0.033323136 |
| ENSMUSG00000035692 | Isg15 | 2.021065991 | 6.53E-05 | 0.034443524 |

**Table 2. Gene Ontology (GO) analysis of genes that are downregulated in the *Capn15*-/- mice.**

| | Mus musculus | Genes downregulated | | | | | |
|---|---|---|---|---|---|---|---|
| GO cellular component complete | # | # | expected | Fold Enrichment | +/- | raw P value | FDR |
| Collagen-containing extracellular matrix | 396 | 7 | .68 | 10.23 | + | 4.55E-06 | 2.27E-03 |
| extracellular matrix | 522 | 8 | .90 | 8.87 | + | 2.50E-06 | 4.98E-03 |
| external encapsulating structure | 524 | 8 | .91 | 8.83 | + | 2.57E-06 | 2.56E-03 |
| extracellular region | 2621 | 16 | 4.53 | 3.53 | + | 2.67E-06 | 1.77E-03 |

[43]. Smarca4, also known as Brg1 is also a putative target for Capn15 cleavage as identified by TAILS (see below). Pax2, Pax5 and Smarca4 all showed increases after loss of Capn 15 (Fig 1C,D,F), and the changes in Pax2 and Pax 5 were significant (Fig. 1D,F). No change was seen with Rnx2 (Fig. 1E). However, we did not detect the loss of any cleavage product of Pax2, Pax 5 or Smarca4 when Capn15 was removed (Fig 1C, D, F).

To identify proteins that might be cleaved by Capn15, we subjected brain homogenates from P2 male and female brains from Capn15 KO and Het mice to an N terminomics/TAILS protocol. Proteins from Capn15 KO mice were labelled with light formaldehyde (+28 Da dimethylation) while proteins from Capn15 Het mice were labelled with heavy formaldehyde (+34 Da dimethylation) (Fig 2A). Prior to the TAILS analysis, the proteomics data from these samples led to the identification of a number of changes (S3 Extended Data Table). Based on the pre-TAILS data, we selected two proteins that are reduced in Capn15 KO mice: Dhx9 and Rbfox2. We did not detect a significant difference in either Dhx9 (Fig 2B) or Rbfox2 (Fig 2C).

The neo-N-termini which represent the cleaved peptides are enriched by removing N-terminals formed after trypsin cleavage using the dendritic polyglycerol aldehyde TAILS polymer. In the TAILS data, we analysed the top 50 peptides (7.3%) that are less abundant in the Capn15 KO brains (S4 Extended Data Table). Most peptides are generated from internal processing (78%), in addition to other cleavage events, such as signal peptide removal (16%) and initial methionine removal (6%) (Fig 3A). As CAPN15 is a protease, we suspected that there would be less neo-N-terminal peptides in the KOs. As a start, we generated IceLogos to determine cleavage site preferences between peptides that are less abundant in Capn15 KO brains and peptides that are not changed. Results from IceLogos showed a preference of P0 lysine and arginine residues in peptides that are less abundant in Capn15 KO samples. However, in peptides that are not changed, arginine is the only preferred residues (Fig 3B). We repeated the experiments by swapping the labels: proteins from Capn15 KO mice were labelled with heavy formaldehyde (+34 Da dimethylation) and proteins from Capn15 heterozygous mice were labelled with light formaldehyde (+28 Da dimethylation). Interestingly, we noticed that the signal from light formaldehyde is much higher than that from heavy formaldehyde leading to many cleavage products apparently being lost in the Capn15 KOs due to the inconsistent labelling. Given that there are more peptides labelled in experiment 2, we analysed the top 100 peptides (7.7%) that are less abundant in the CAPN15 KO brains. Similar to experiment 1, internal processing is the predominant form of cleavage events (77%; Fig 3A). Results from IceLogos showed no cleavage site preference between the two groups (Fig 3B). While lysine residues were not observed in the IceLogo of experiment 2, this may be due to the large number of peptides lost in Capn15 due to differences in labelling efficiency. While there are a number of recent computational approaches to determine calpain cleavage sites [44–46], these use known calpain sites to generate their algorithms and since no CAPN15 sites have yet been positively identified, it is problematic to use a computational approach at this time. None of the putative sites identified here are predicted as calpain sites by these algorithms.

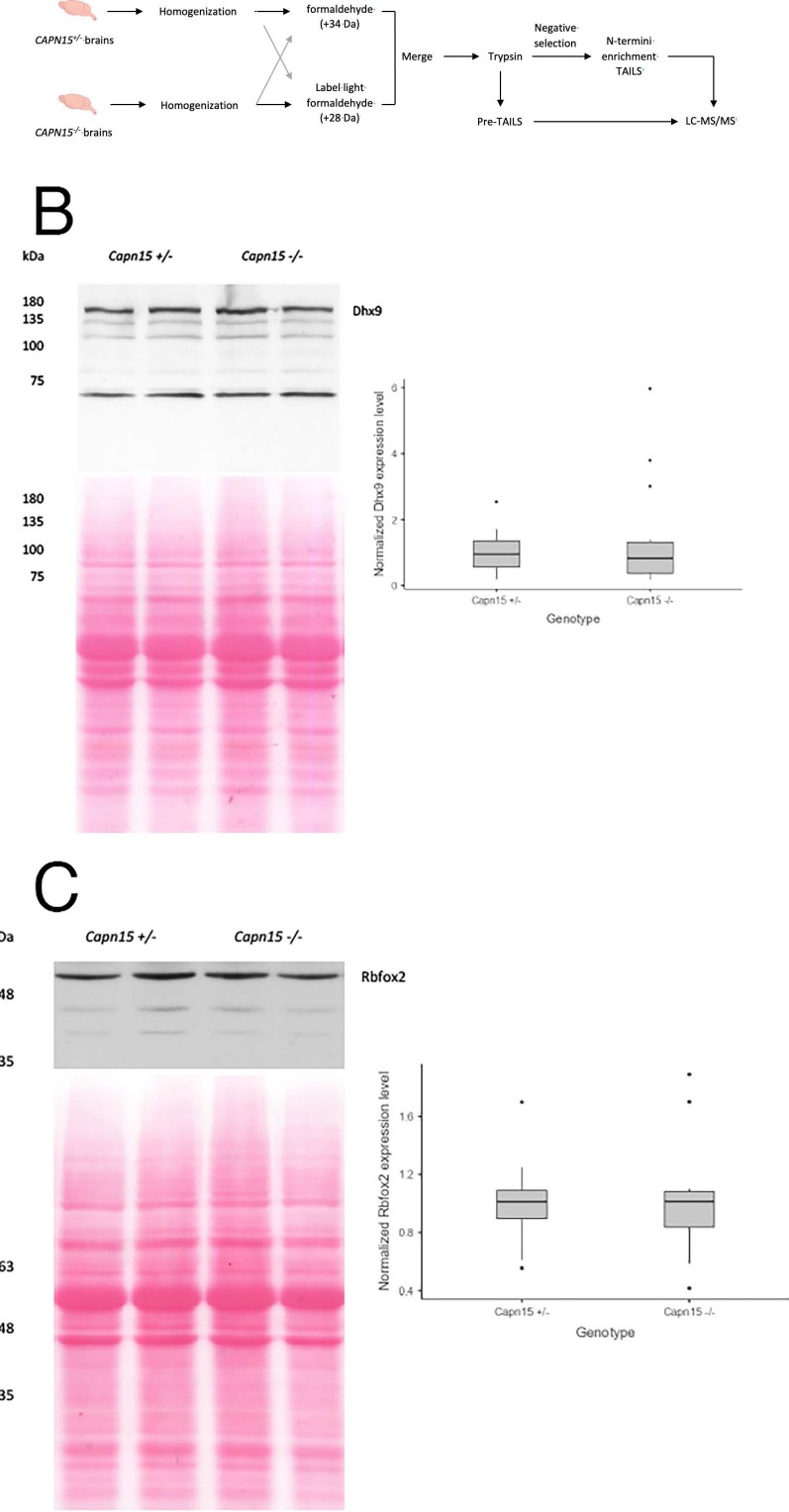

**Fig 2. Proteomics analyses of brain homogenates from *CAPN15* +/- and *CAPN15*-/- mice.** (A) Workflow of pro-
teomics experimental design. In experiment 1 (n = 3), peptides from *Capn15*+/- mice are labeled with heavy formalde-
hyde and peptides from *Capn15*-/- mice are labeled with light formaldehyde. In experiment 2 (n = 5), the experiment
was repeated with peptides from *Capn15*-/- mice being labeled with heavy formaldehyde. (B) Left, Western blot of

P2 mouse brain homogenate targeting Dhx9. Right, Box and whisker plot showing the normalized quantification of Dhx9 in *Capn15-/-* mice (t-test, $p = 0.38$). (C) Left, Western blot of P2 mouse brain homogenate targeting Rbfox2. Right, Box and whisker plot showing the normalized quantification of Rbfox2 in *Capn15-/-* KO mice (t-test, $p = 0.95$). n = 13 for *Capn15+/-* mice; n = 14 for *Capn15-/-* mice. Dots are data points outside the box and whiskers.

Due to the role of CAPN15 in neurodevelopment, we selected several candidate proteins from TAILS data that are important to neurodevelopment and are cleaved after either arginine or lysine (Table 3). Ctnnb1, Tubb3, eEF2, Crmp1, and histone H2A are all cleaved after arginine. We did not detect any significant changes in their expression level (Fig 4A-D) except for Tubb3, whose expression is significantly higher in the Capn15 KO mice ($p = 0.014$, Fig 4E). However, no cleavage product was observed. We also looked at two candidates, histone H4 and doublecortin Dcx, which have peptides that are cleaved after lysine residues. While we did not detect a significant change in H4 expression (Fig 4F), we found that the expression of Dcx is significantly higher in Capn15 KO mice (Fig 4G). According to TAILS data, Dcx is cleaved between residues $^{151}K \downarrow T^{152}$. Interestingly, we observed a band that is slightly bigger than 17kDa, which matches the size of the N-terminal cleavage product of Dcx. However, the expression of this cleavage product is significantly higher is Capn15 KO mice as well ($p = 0.044$, Fig 4G) inconsistent with this product being due to cleavage by Capn15.

## Discussion

Loss of *Capn15* in mice or homozygous mutations in *CAPN15* in humans have some strong effects on development and neurodevelopment in particular [7,8,10]. In this study, we attempted to identify putative Capn15 substrates through RNA sequencing and N-terminomics. There has been a great deal of work on determining the cleavage site of the classical Calpains (Calpain-1 and -2), but even with 100s of known substrates, it requires even higher numbers to better predict the ideal site of cleavage which often are in a disordered domain [46]. There is virtually no information on sites cleaved by non-classical calpains, although one substrate has been determined for the Palb Calpain-7 and this has a lysine at the P1 position [47] consistent with the apparent preference shown by Capn15. It has been shown that Calpain-10 is responsible for the separation of MAP1B heavy and light chain, which is cleaved after a serine residue [48]. Calpain-10 is an atypical calpain with two catalytic domains, but how it derived from other calpains is not clear [1].

TAILS works by first blocking via dimethylating the primary amines of endogenous N-terminals and lysines. Internal peptides created by the subsequent trypsin treatment are removed by binding to a high molecular weight, aldehyde derived polyglycerol polymer (HPG-ALD) followed by filtration and then mass spectrometry of the endogenous N-terminal peptides remaining [49]. While trypsin cleaves after lysines and arginines, the majority of background peptides found in these experiments begin with an arginine as lysine dimethylation prevents trypsin cleavage [49]. Thus, background from peptides that start with arginine are mainly due to incomplete removal by HPG-ALD, while background from peptides that start with lysine is due to both incomplete demethylation and incomplete removal by HPG-ALD. Thus, most background peptides in these studies are cleaved after arginine. Since peptides starting with lysine were only seen in the IceLogo of reduced peptides, it is likely that this represents a "true" preference of Capn15 for basic residues. However, the lack of detection of the loss of any cleaved peptide on immunoblots tempers any confidence in this finding.

When we observed significant changes due to the loss of Capn15 (Tubb3, Pax2, Pax5, Doublecortin), it was due to an increase in total levels of the proteins. Capn15 binds poly-ubuqiutin and it is possible that Capn15 cleavage is followed quickly by proteosome mediated loss of the cleavage products. Capn15 loss is the cause of a small subset of patients with

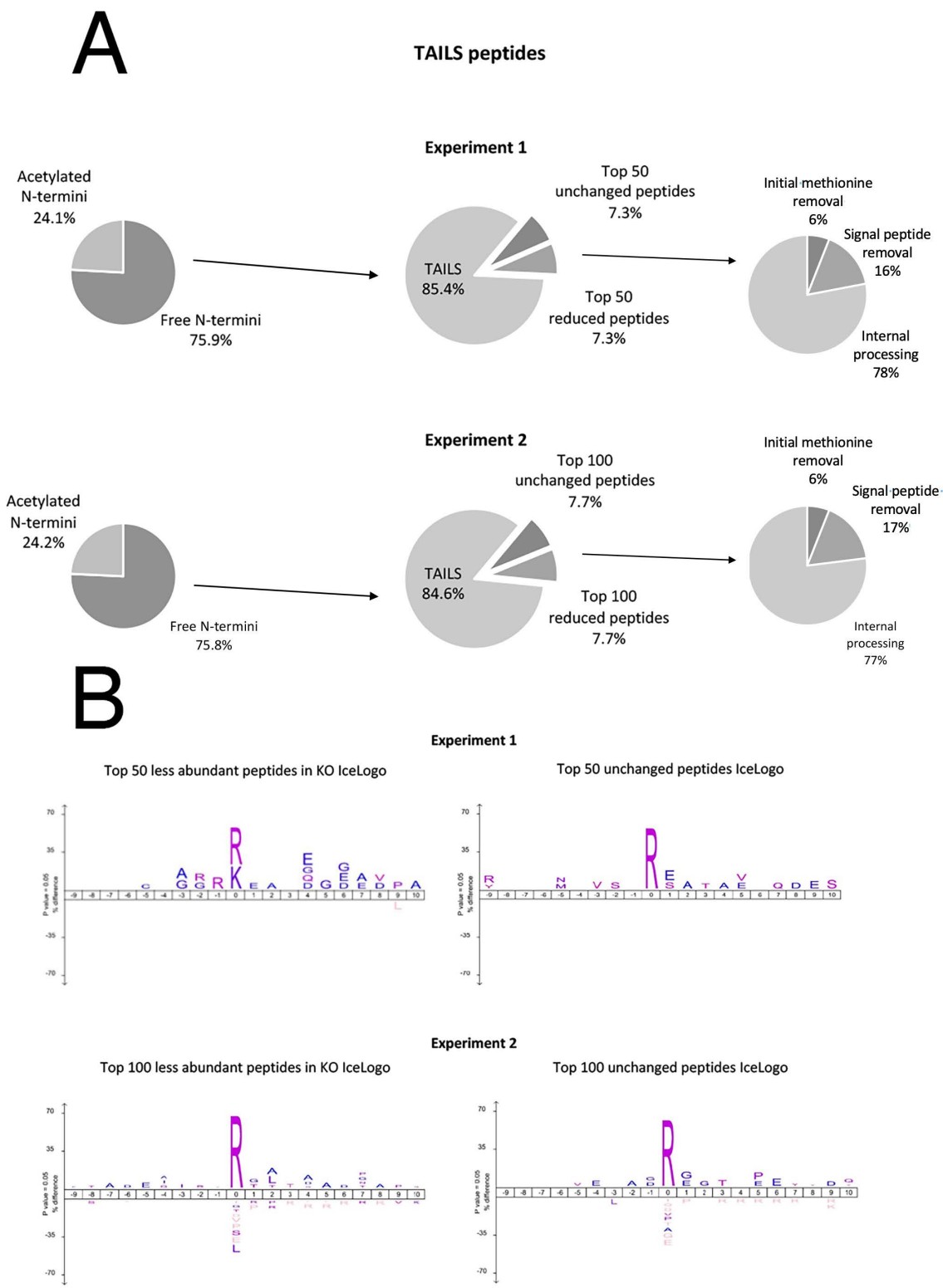

**Fig 3. N-terminomics/TAILS analyses of brain homogenates from *Capn15* +/- and *Capn15*-/- mice.** (A)Top, analyses of peptides from experiment 1. Left, distribution of N-terminal peptides in the TAILS enrichment. Middle, top 50 reduced and unchanged neo-N-terminal peptides in brain homogenate from *Capn15*-/- mice are selected for analyses. Right, distribution of post-translational peptide modifications as analyzed using TopFINDER in top 50 reduced peptides. Bottom, analyses of peptides from experiment 2. Left, distribution of N-terminal peptides in the TAILS enrichment. Middle, top 100 reduced and

unchanged neo-N-terminal peptides in brain homogenate from *Capn15-/-* mice are selected for analyses. Right, distribution of post-translational peptide modifications as analyzed using TopFINDER in top 50 reduced peptides. (B) Peptide sequences surrounding the cleavage site are found using TopFINDer. Top, analyses of peptides from experiment 1. Left, peptide sequence profiles of top 50 reduced neo-N-terminal peptides in brain homogenate from *Capn15-/-* mice identified in the TAILS analysis using IceLogo. Right, peptide sequence profiles of top 50 unchanged neo-N-terminal peptides in brain homogenate from *Capn15-/-* mice identified in the TAILS analysis using IceLogo. Bottom, analyses of peptides from experiment 2. Left, peptide sequence profiles of top 100 reduced neo-N-terminal peptides in brain homogenate from *Capn15-/-* mice identified in the TAILS analysis using IceLogo. Right, peptide sequence profiles of top 100 unchanged neo-N-terminal peptides in brain homogenate from *Capn15-/-* mice identified in the TAILS analysis using IceLogo. Peptides are cleaved between P0 and P1.

Johanson-Blizzard syndrome [8]. This syndrome is mainly caused by the loss of the ubiquitin ligase UBR1, part of the N-end rule ubiquitin pathway that leads to degradation of proteins starting with a basic residue. However, we do not see an excess of lysine and arginine at the P1 position as would be expected if these two enzymes worked in parallel.

## Putative substrates

While we cannot validate any changes in Enpp2 level, as suggested by RNA-SEQ, we did observe an increase in Pax2 and Pax 5 levels in the KOs. Pax2 is a transcription factor that are critical during early eye development and is both spatially and temporally regulated. During eye development, Pax2 is restricted to the ventral optic cup and later downregulated for further eye development. A failure in downregulating Pax2 leads to colobomas [50]. Pax5 is a recent gene duplication of Pax2, and while biologically equivalent, is expressed in different regions of the brain than Pax5 [51] Our results suggest that Capn15 might play an important role during early development by regulating the level of transcription factors, either through cleavage followed by degradation or perhaps indirectly though some unidentified Capn15 substrates.

We discovered that DCX might be a putative cleavage product of Capn15. The cleavage site is located in between the two N-terminal doublecortin domains. Currently, Dcx cleavage is not documented in the literature. DCLK2 however, a protein with the doublecortin domain, can be cleaved by Calpain-2 [52]. Other than DCX and DCLK2, another microtubule interacting proteins, MAP1B, can also be cleaved by calpain. Interestingly several peptides of MAP1B are identified by TAILS, although none of them overlaps the Calpain-10 cleavage site [48]. The two proteins identified in the TAILS as putatively substrates of Capn15 that are significantly increased in the *Capn15* KO mice, Tubb3 and Dcx, are both associated to the microtubule system. Mutations in microtubule proteins are usually associated to malformation of cortical development, and sometimes cerebellar development, which was observed in some of the patients with variations in *CAPN15* gene as well [10,53]. While tubulinopathies can be explained by haploinsufficiency and dominant negativity [54], it is unclear whether an increase in expression can have similar effects.

## Strengths and limitations of this study

This study has several strengths. It is the first examination of the effects of loss of Capn 15 on mRNA, protein levels, and cleavage products (TAILs) in any system and a number of significant changes were observed. Several of these were followed up by immunoblotting and significant changes in putative transcription factors that mediated the changes in RNA-SEQ were identified. This leads to strong hypotheses for future studies. There are limitations in this study. We focused on P2 brains as Capn15 is widely expressed at this time, but levels of Capn15 are higher in embryonic brain and many proteins whose loss is responsible for neurodevelopmental disorders act earlier during neuronal differentiation that peaks in embryonic

**Table 3. TAILS data of peptides that are validated in the experiments.**

| Sequence | Gene Names | Ratio H/L normalized | P10 to P1 | P1' to P10' |
|---|---|---|---|---|
| DSEAGSSTPTTSTR | Smarca4 | 2.6621 | QKKSSRKRKR | DSEAGSSTPT |
| | | NaN | | |
| HAVVNLINYQDDAELATR | Ctnnb1 | 0.45535 | RLAEPSQMLK | HAVVNLINYQ |
| | | NF | | |
| TMQNTNDVETAR | Ctnnb1 | 0.20683 | SPQMVSAIVR | TMQNTNDVET |
| | | NaN | | |
| HQEAEMAQNAVR | Ctnnb1 | 0.10106 | ICALRHLTSR | HQEAEMAQNAV |
| | | 0.030254 | | |
| SLGGGTGSGMGTLLISKVR | Tubb3 | 1.0187 | DCLQGFQLTH | SLGGGTGSGM |
| | | 0.0062467 | | |
| LHFFMPGFAPLTAR | Tubb3 | 0.19522 | LAVNMVPFPR | LHFFMPGFAP |
| | | 0.077518 | | |
| CLYASVLTAQPR | eEF2 | 0.1897 | GGQIIPTARR | CLYASVLTAQ |
| | | 0.028074 | | |
| GGGQIIPTAR | eEF2 | 0.020022 | VTLHADAIHR | GGGQIIPTAR |
| | | NaN | | |
| QIGENLIVPGGVKTIEANGR | Crmp1 | 1.7041 | DVYLEDGLIK | QIGENLIVPG |
| | | NF | | |
| NLHQSNFSLSGAQIDDNNPR | Crmp1 | 0.64022 | PSKHQPPPIR | NLHQSNFSLS |
| | | 0.032116 | | |
| DNFTLIPEGVNGIEER | Crmp1 | 0.22991 | YSTAQKAVGK | DNFTLIPEGV |
| | | 0.16699 | | |
| ALSRPEELEAEAVFR | Crmp1 | 0.056725 | EMGITGPEGH | ALSRPEELEA |
| | | 0.01421 | | |
| AGLQFPVGR | H2A | 0.022466 | RAKAKSRSSR | AGLQFPVGRV |
| | | 0.0010542 | | |
| DNIQGITKPAIR | H4 | 1.2121 | GAKRHRKVLR | DNIQGITKPA |
| | | 0.00022067 | | |
| DAVTYTEHAKR | H4 | 0.73879 | LKVFLENVIR | DAVTYTEHAK |
| | | NaN | | |
| GVLKVFLENVIR | H4 | 0.98424 | ISGLIYEETR | GVLKVFLENV |
| | | 0.11 | | |
| ISGLIYEETR | H4 | 0.28817 | RLARRGGVKR | ISGLIYEETR |
| | | 0.00086128 | | |
| KTVTAMDVVYALKR | H4 | 0.74607 | AVTYTEHAKR | KTVTAMDVVY |
| | | 0.0209 | | |
| TVTAMDVVYALKR | H4 | 3.1745 | VTYTEHAKRK | TVTAMDVVYA |
| | | NaN | | |
| TSANMKAPQSLASSNSAQAR | Dcx | 1.3391 | VNPNWSVNVK | TSANMKAPQS |
| | | NaN | | |
| APQSLASSNSAQAR | Dcx | 0.32346 | VNVKTSANMK | APQSLASSNS |
| | | 0.010968 | | |
| SLSDNINLPQGVR | Dcx | 0.050876 | SFDALLADLTR | SLSDNINLPQ |
| | | NaN | | |

Peptides from experiment 1 are highlighted in green and the ratio H/L of the same peptides from experiment 2 are highlighted in grey. NaN, peptide with either of the label is not detected in TAILS data. NF, peptide is not found in TAILS data.

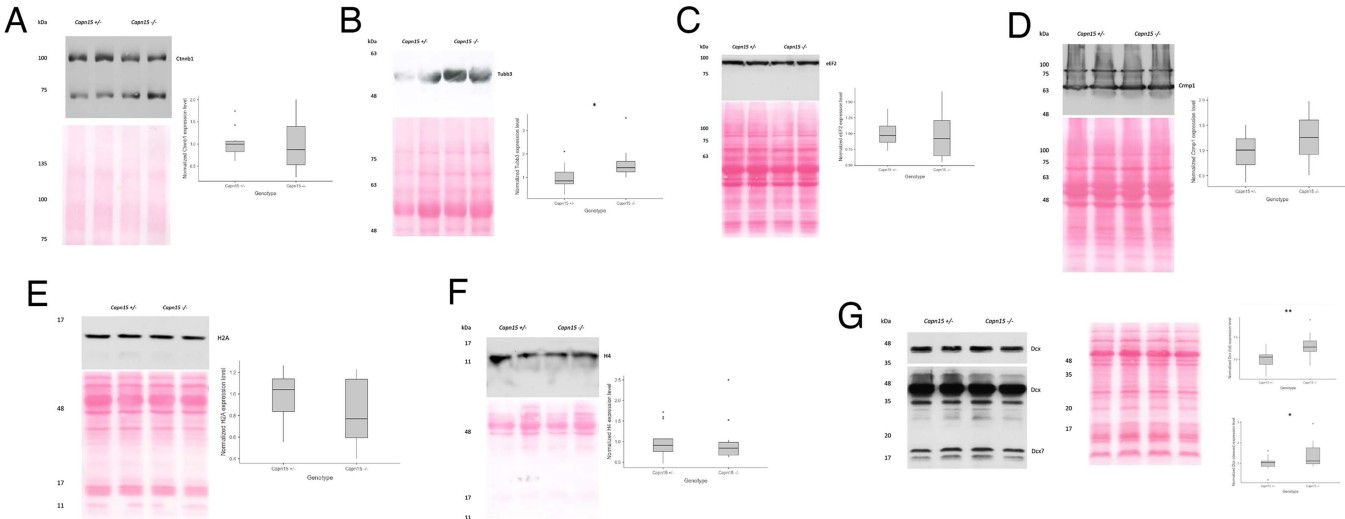

**Fig 4. Verification of TAILS targets with western blots.** (A) Left, Western blot of P2 mouse brain homogenate targeting Ctnb1. Right, Box and whisker plot showing the normalized quantification of Ctnnb1 in *Capn15-/-* mice (t-test, $p = 0.98$). (B) Left, Western blot of P2 mouse brain homogenate targeting Tubb3. Right, Box and whisker plot showing the normalized quantification of Tubb3 in *Capn15-/-* mice (t-test, $p = 0.95$). (C) Left, Western blot of P2 mouse brain homogenate targeting eEF2. Right, Box and whisker plot showing the normalized quantification of eEF2 in *Capn15-/-* mice (t-test, $p = 0.69$). (D) Left, western blot of P2 mouse brain homogenate targeting Crmp1. Right, Box and whisker plot showing the normalized quantification of Crmp1 in *Capn15-/-* mice (t-test, $p = 0.19$). (E) Left, western blot of P2 moues brain homogenate targeting H2A. Right, Box and whisker plot showing the normalized quantification of H2A in *Capn15-/-* mice (t-test, $p = 0.12$). (F) Left, Western blot of P2 mouse brain homogenate targeting H4. Right, Box and whisker plot showing the normalized quantification of H4 in *Capn15-/-* mice (t-test, $p = 0.83$). (G) Left, Western blot of P2 moues brain homogenate targeting Dcx. Right, Box and whisker plot showing the normalized quantification of full length Dcx (t-test, $p = 0.002$) and putative cleavage product of Dcx in *Capn15-/-* mice (t-test, $p = 0.044$). n = 13 for *Capn15+/-* mice; n = 14 for *Capn15-/-* mice. Dots are data points outside the box and whiskers. $*p < 0.05$, $**p < 0.01$.

brains. Capn15 may also be important in selected areas (such as the primordial eye fields) and using whole brains may dilute possible changes. It will be important in the future to determine if increases in the level of the proteins identified to be changed in Capn 15 Kos (Pax2, Pax5, Tubb3 and Dcx) play a role in the phenotypes seen in this KO mouse. While the use of heterozygotes of wild type allowed for using littermates in the TAILS assay, it may have reduced the amount of changes in the affected peptides.

## Conclusions

We have used several technologies to identify putative substrates of Capn15 in P2 mouse brains. While several proteins showed a significant increase in their total levels, we did not identify any cleavage product that is lost. We still believe, based on the conservation of the Sol calpain catalytic domain residues over evolution, that this family of calpains is a protease, but further work will be required to verify this and to identify its substrates. While the increase in peptides after lysines in the TAILS assay is intriguing, it is not yet definitive evidence for the substrate specificity of Capn15.

## Supporting information

**S1 Supplemental Data. Full size images of all blots used in quantification.**
(PDF)

**S2 Supplemental Data. LogCPM values for every mRNA for all samples used in RNA-SEQ experiments.**
(CSV)

**S1 Extended DataTable. LISA analyses of differentially expressed genes.** Transcription factors that are predicted to regulate differentially expressed genes based on consensus sequence of the binding motif.
(XLSX)

**S2 Extended Data Table. LISA analyses of differentially expressed genes.** Transcription factors that are predicted to regulate differentially expressed genes based on ChIP-seq data.
(XLSX)

**S3 Extended Data Table. Pre-enrichment TAILS protein from CAPN15 Het and KO mice.** Data are selected within two standard deviation of log2 of normalized ratio H/L to exclude extreme values (entries highlighted in green). Rbfox2 and Dhx9 are highlighted in yellow. Entries with 0 total intensity are filtered.
(XLSX)

**S4 Extended Data Table. TAILS peptides from CAPN15 Het and KO mice.** Top 50 entries of the highest normalized ratio H/L are selected as the top 50 reduced peptides (highlighted in green). Top 50 entries around the average of normalized ratio H/L are selected as the top 50 unchanged peptides (highlighted in beige). Entries with 0 total intensity are filtered.
(XLSX)

## Author contributions

**Conceptualization:** Congyao Zha, Antoine Dufour, Wayne S. Sossin.

**Formal analysis:** Congyao Zha, Senthilkumar Kailasam, Daniel Young, Wayne S. Sossin.

**Funding acquisition:** Wayne S. Sossin.

**Investigation:** Congyao Zha, Ally Huang, Senthilkumar Kailasam, Daniel Young, Antoine Dufour.

**Methodology:** Senthilkumar Kailasam, Daniel Young, Antoine Dufour.

**Resources:** Wayne S. Sossin.

**Software:** Senthilkumar Kailasam, Daniel Young.

**Supervision:** Antoine Dufour, Wayne S. Sossin.

**Validation:** Congyao Zha, Ally Huang.

**Visualization:** Congyao Zha.

**Writing – original draft:** Congyao Zha, Antoine Dufour, Wayne S. Sossin.

**Writing – review & editing:** Congyao Zha.

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
