## [Decision Letter · Decision Letter 0]

7 Oct 2024

PONE-D-24-18366Identifying putative substrates of Calpain-15 in neurodevelopmentPLOS ONE

Dear Dr. Sossin,

Thank you for submitting your manuscript to PLOS ONE. After careful consideration, we feel that it has merit but does not fully meet PLOS ONE’s publication criteria as it currently stands. Therefore, we invite you to submit a revised version of the manuscript that addresses the points raised during the review process.

We look forward to receiving your revised manuscript.

Kind regards,

Aldrin V. Gomes, Ph.D.

Academic Editor

PLOS ONE

Journal Requirements:

https://doi.org/10.1016/j.bbr.2023.114635

In your revision ensure you cite all your sources (including your own works), and quote or rephrase any duplicated text outside the methods section. Further consideration is dependent on these concerns being addressed.

5. Please expand the acronym “CIHR” (as indicated in your financial disclosure) so that it states the name of your funders in full.

6. Thank you for stating the following financial disclosure: “CIHR grant 340328 to WSS”

7. Please note that funding information should not appear in the Acknowledgments section or other areas of your manuscript. We will only publish funding information present in the Funding Statement section of the online submission form. Please remove any funding-related text from the manuscript. 

8. Please note that your Data Availability Statement is currently missing the repository name. If your manuscript is accepted for publication, you will be asked to provide these details on a very short timeline. We therefore suggest that you provide this information now, though we will not hold up the peer review process if you are unable.

9. PLOS ONE now requires that authors provide the original uncropped and unadjusted images underlying all blot or gel results reported in a submission’s figures or Supporting Information files. This policy and the journal’s other requirements for blot/gel reporting and figure preparation are described in detail at https://journals.plos.org/plosone/s/figures#loc-blot-and-gel-reporting-requirements and https://journals.plos.org/plosone/s/figures#loc-preparing-figures-from-image-files. When you submit your revised manuscript, please ensure that your figures adhere fully to these guidelines and provide the original underlying images for all blot or gel data reported in your submission. See the following link for instructions on providing the original image data: https://journals.plos.org/plosone/s/figures#loc-original-images-for-blots-and-gels.   

10. Please amend the manuscript submission data (via Edit Submission) to include author “Ally Huang”. 

**Additional Editor Comments:**

While both reviewers commented favorably on your manuscript, some concerns need to be addressed. These include validation that the different sex of the mice did not affect the results and validation of more hits. The authors do not need to address the suggestion that other ages or specific regions of the brain should be investigated (concern #5 of one of the reviewers).

Reviewers' comments:

Reviewer's Responses to Questions

**Comments to the Author**

1. Is the manuscript technically sound, and do the data support the conclusions?

Reviewer #1: Yes

Reviewer #2: Yes

2. Has the statistical analysis been performed appropriately and rigorously? 

Reviewer #1: Yes

Reviewer #2: Yes

3. Have the authors made all data underlying the findings in their manuscript fully available?

Reviewer #1: Yes

Reviewer #2: Yes

4. Is the manuscript presented in an intelligible fashion and written in standard English?

Reviewer #1: No

Reviewer #2: Yes

5. Review Comments to the Author

Reviewer #1: I think this research paper is technically sound and appropriately analyzed. The one minor issue I noticed during my read-through, I noticed on line 338, that the paper says "moues brain" instead of "mouse brain."

Reviewer #2: This manuscript by Zha et al. examined the putative substrates of Calpain 15 (Capn15) by using Capn15 KO P2 mice of both sexes. The authors used RNA-seq proteomics, and N-terminomics/terminal amino isotopic labelling of substrates (TAILS) to elucidate the probable substrates of capn15. This manuscript aimed to provides some insights into the unidentified Capn 15 substrates. However, the manuscript can be strengthened by further addressing the points below.

1. The authors stated using both sexes of CapN15 KO mice. While it might be difficult to correctly identify the mice sex at P2, does the author have any evidence that there are no sex differences in these KO mice and if that influences the 38 mRNAs found to have significant differential gene expression changes? Please address this possibility.

2. Figure 1A and 1B: It is definitely critical to represent the up- and down-regulated genes in a figure rather than a table since it is being shown as a figure. The current version can be included as a table by itself.

3. The authors tested only two of the hits from the LISA analysis screen: Pax and Smarca4. Pax2 was shown to be significantly increased. I suggest the authors test and validate more than those two. Furthermore, additional work such as investigating effect of Pax2 inhibition can be performed. The same can be performed for Dcx that was suggested to be a putative cleavage product of Capn15.

4. The study will benefit from employing a computational approach that can help in labeling calpain substrate cleavage sites from amino acid sequence using conditional random fields or any computational technique using the sequences of peptides obtained from the TAILS experiment.

5. Since no substrate was validated and found in this study I suggest using other ages and not just P2 as well as focusing on a particular area of the brain such as cerebral cortex as opposed to using whole brain.

6. I believe this current manuscript can be really strengthened by performing further in vivo validation experiments.

Minor:

7. Figure 4: The images in figure 4 are not well aligned and seem scattered around.

6. PLOS authors have the option to publish the peer review history of their article (what does this mean? ). If published, this will include your full peer review and any attached files.

**Do you want your identity to be public for this peer review?** For information about this choice, including consent withdrawal, please see our Privacy Policy .

Reviewer #1: No

Reviewer #2: No

---

## [Author Response · Author response to Decision Letter 1]

21 Nov 2024

We have added a reponse to reviewer section as a file.

---

## [Decision Letter · Decision Letter 1]

12 Jan 2025

PONE-D-24-18366R1Identifying putative substrates of Calpain-15 in neurodevelopmentPLOS ONE

Dear Dr. Sossin,

Thank you for submitting your manuscript to PLOS ONE. After careful consideration, we feel that it has merit but does not fully meet PLOS ONE’s publication criteria as it currently stands. Therefore, we invite you to submit a revised version of the manuscript that addresses the points raised during the review process.

The manuscript has been evaluated by two reviewers, and their comments are available below. There are some remaining requests for the underlying data and further discussion. Could you please carefully revise the manuscript to address all comments raised?

We look forward to receiving your revised manuscript.

Kind regards,

Helen Howard

Staff Editor

PLOS ONE

Journal Requirements:

Additional Editor Comments (if provided):

Reviewers' comments:

Reviewer's Responses to Questions

**Comments to the Author**

1. If the authors have adequately addressed your comments raised in a previous round of review and you feel that this manuscript is now acceptable for publication, you may indicate that here to bypass the “Comments to the Author” section, enter your conflict of interest statement in the “Confidential to Editor” section, and submit your "Accept" recommendation.

Reviewer #1: All comments have been addressed

Reviewer #2: All comments have been addressed

2. Is the manuscript technically sound, and do the data support the conclusions?

Reviewer #1: Yes

Reviewer #2: Yes

3. Has the statistical analysis been performed appropriately and rigorously? 

Reviewer #1: Yes

Reviewer #2: Yes

4. Have the authors made all data underlying the findings in their manuscript fully available?

Reviewer #1: Yes

Reviewer #2: Yes

5. Is the manuscript presented in an intelligible fashion and written in standard English?

Reviewer #1: Yes

Reviewer #2: Yes

6. Review Comments to the Author

Reviewer #1: All comments to the author has been addressed. I believe that the manuscript is technically sound and the data does support the conclusion.

Reviewer #2: The authors have partly addressed comments as it seems more work would be needed to strengthen this study.

1. However, I think it's advisable for the authors to make the raw and processed RNA-seq data available by depositing in the GEO database. The proteomics data should also be deposited in a public repository database such as Global Proteome Machine Database (GPMDB), PeptideAtlas, the PRoteomics IDEntifications database (PRIDE), Tranche, and NCBI Peptidome.

2. Additionally, the authors should discuss the strengths and limitations of the analysis from their study.

7. PLOS authors have the option to publish the peer review history of their article (what does this mean? ). If published, this will include your full peer review and any attached files.

**Do you want your identity to be public for this peer review?** For information about this choice, including consent withdrawal, please see our Privacy Policy .

Reviewer #1: No

Reviewer #2: No

---

## [Author Response · Author response to Decision Letter 2]

24 Jan 2025

The response to reviews was added as a PDF. The contents of this file are reproduced below.

Response to Reviewers

Dear Editor,

Thank you for the reviews. Below are the response to the reviewers. There were no comments from the academic editor. We hope the paper is now acceptable for publication at PLOS One.

Reviewer #1: All comments to the author has been addressed. I believe that the manuscript is technically sound and the data does support the conclusion.

Thank you for your support.

Reviewer #2: The authors have partly addressed comments as it seems more work would be needed to strengthen this study.

1. However, I think it's advisable for the authors to make the raw and processed RNA-seq data available by depositing in the GEO database. The proteomics data should also be deposited in a public repository database such as Global Proteome Machine Database (GPMDB), PeptideAtlas, the PRoteomics IDEntifications database (PRIDE), Tranche, and NCBI Peptidome.

We have deposited the RNA-SEQ data in the GEO database. We have also attached the full RNA-SEQ analysis as a supplemental data file. We have uploaded the proteomics data to the PRIDE database. The accession numbers have been added to the methods section.

If the reviewer wishes to examine these deposits, information is below. Data will be released to the public once we have the paper DOI information.

Reviewer access details

Log in to the PRIDE website using the following details:

Project accession: PXD060034

Token: qm4WU3lBbgV5

----------

To review GEO accession GSE287980:

Go to https://can01.safelinks.protection.outlook.com/?url=https%3A%2F%2Fwww.ncbi.nlm.nih.gov%2Fgeo%2Fquery%2Facc.cgi%3Facc%3DGSE287980&data=05%7C02%7Csenthil.duraikannukailasam%40mcgill.ca%7C65e4b9ef41ce4920189908dd3ca2fd8a%7Ccd31967152e74a68afa9fcf8f89f09ea%7C0%7C0%7C638733392919091956%7CUnknown%7CTWFpbGZsb3d8eyJFbXB0eU1hcGkiOnRydWUsIlYiOiIwLjAuMDAwMCIsIlAiOiJXaW4zMiIsIkFOIjoiTWFpbCIsIldUIjoyfQ%3D%3D%7C0%7C%7C%7C&sdata=I%2FkREJacdGS%2BBRt%2B3tnkLMLMwQENAkKQeRMrJF20zEA%3D&reserved=0

Enter token apwzwkmitbudtkz into the box

2. Additionally, the authors should discuss the strengths and limitations of the analysis from their study.

We had a section on the limitations of the study in the discussion, but it was not labeled. We have expanded this section and added strengths as well. This section of the discussion is reproduced below.

Strengths and Limitations of this study

This study has several strengths. It is the first examination of the effects of loss of Capn 15 on mRNA, protein levels, and cleavage products (TAILs) in any system and a number of significant changes were observed. Several of these were followed up by immunoblotting and significant changes in putative transcription factors that mediated the changes in RNA-SEQ were identified. This leads to strong hypotheses for future studies. There are limitations in this study. We focused on P2 brains as Capn15 is widely expressed at this time, but levels of Capn15 are higher in embryonic brain and many proteins whose loss is responsible for neurodevelopmental disorders act earlier during neuronal differentiation that peaks in embryonic brains. Capn15 may also be important in selected areas (such as the primordial eye fields) and using whole brains may dilute possible changes. It will be important in the future to determine if increases in the level of the proteins identified to be changed in Capn 15 Kos (Pax2, Pax5, Tubb3 and Dcx) play a role in the phenotypes seen in this KO mouse. While the use of heterozygotes of wild type allowed for using littermates in the TAILS assay, it may have reduced the amount of changes in the affected peptides.

---

## [Editor Report · Decision Letter 2]

4 Feb 2025

Identifying putative substrates of Calpain-15 in neurodevelopment

PONE-D-24-18366R2

Dear Dr. Sossin,

We’re pleased to inform you that your manuscript has been judged scientifically suitable for publication and will be formally accepted for publication once it meets all outstanding technical requirements.

Kind regards,

Patrick Goymer

Staff Editor

PLOS ONE
---

## [Editor Report · Acceptance letter]

PONE-D-24-18366R2

PLOS ONE

Dear Dr. Sossin,

I'm pleased to inform you that your manuscript has been deemed suitable for publication in PLOS ONE. Congratulations! Your manuscript is now being handed over to our production team.

Kind regards,

on behalf of

Dr Patrick Goymer

Staff Editor

PLOS ONE